# Facilitating Drug Discovery in Breast Cancer by Virtually Screening Patients Using In Vitro Drug Response Modeling

**DOI:** 10.3390/cancers13040885

**Published:** 2021-02-20

**Authors:** Robert F. Gruener, Alexander Ling, Ya-Fang Chang, Gladys Morrison, Paul Geeleher, Geoffrey L. Greene, R. Stephanie Huang

**Affiliations:** 1Ben May Department for Cancer Research, University of Chicago, Chicago, IL 60637, USA; rgruener@uchicago.edu (R.F.G.); g39220007@gmail.com (Y.-F.C.); ggreene@uchicago.edu (G.L.G.); 2Department of Experimental and Clinical Pharmacology, University of Minnesota, Minneapolis, MN 55455, USA; alexander.l.ling@gmail.com; 3Committee for Clinical Pharmacology and Pharmacogenomics, University of Chicago, Chicago, IL 60637, USA; gdmorriso@gmail.com; 4Department of Computational Biology, St. Jude Children’s Research Hospital, Memphis, TN 38105, USA; paul.geeleher@stjude.org

**Keywords:** transcriptome, drug discovery, breast cancer, Wee1, drug imputation

## Abstract

**Simple Summary:**

While patient datasets such as The Cancer Genome Atlas (TCGA) often contain a plethora of “-omics” data, the corresponding drug response information are limited and not suited for novel drug discovery. By integrating in vitro high throughput drug screening data and patient tumor molecular information, we created a virtual drug screening pipeline that enables drug discovery with simultaneous biomarker identification for a patient population. Using triple-negative breast cancer (TNBC) as our population of interest, we demonstrated the pipeline from lead identification, to biomarker discovery, to in vitro and in vivo validation of the compound AZD-1775.

**Abstract:**

(1) Background: Drug imputation methods often aim to translate in vitro drug response to in vivo drug efficacy predictions. While commonly used in retrospective analyses, our aim is to investigate the use of drug prediction methods for the generation of novel drug discovery hypotheses. Triple-negative breast cancer (TNBC) is a severe clinical challenge in need of new therapies. (2) Methods: We used an established machine learning approach to build models of drug response based on cell line transcriptome data, which we then applied to patient tumor data to obtain predicted sensitivity scores for hundreds of drugs in over 1000 breast cancer patients. We then examined the relationships between predicted drug response and patient clinical features. (3) Results: Our analysis recapitulated several suspected vulnerabilities in TNBC and identified a number of compounds-of-interest. AZD-1775, a Wee1 inhibitor, was predicted to have preferential activity in TNBC (*p* < 2.2 × 10^−16^) and its efficacy was highly associated with *TP53* mutations (*p* = 1.2 × 10^−46^). We validated these findings using independent cell line screening data and pathway analysis. Additionally, co-administration of AZD-1775 with standard-of-care paclitaxel was able to inhibit tumor growth (*p* < 0.05) and increase survival (*p* < 0.01) in a xenograft mouse model of TNBC. (4) Conclusions: Overall, this study provides a framework to turn any cancer transcriptomic dataset into a dataset for drug discovery. Using this framework, one can quickly generate meaningful drug discovery hypotheses for a cancer population of interest.

## 1. Introduction

Triple-negative breast cancer (TNBC) is a clinically defined subset of breast cancers that are inherently difficult to treat, as unlike the other breast cancer subtypes, they are defined by the absence of a distinct molecular target. Despite some recent progress in the field, TNBC patients continue to have the worst 5-year overall survival among breast cancer patients, and most TNBC patients are still treated with cytotoxic chemotherapies [1]. There is a clear and present need to identify new and effective therapies for TNBC to help reduce morbidity and mortality in these patients.

Traditional drug development pipelines are time-consuming and take years for target identification, validation, and subsequent design optimization of the lead candidate compounds [2]. While these approaches are indispensable for generating new therapeutic compounds, methods are needed to holistically explore and expand the potential use of existing drugs to different cancer contexts. The high costs, low success rates, and protracted development time for establishing new clinically-viable compounds has generated interest in expanding the use of (utility extension) and finding new uses for (repurposing/repositioning) existing drugs [3,4]. The challenge that remains is to identify appropriate contexts for drug repositioning and utility extension. Pathway mapping and signature-based approaches are both examples that utilize gene/protein expression patterns to identify such opportunities [5]. In vitro screening is another common approach to test existing drugs for phenotypic changes in cancer cell lines [6].

In the area of precision medicine, these in vitro screening data are used as the inputs in machine learning models that aim to obtain accurate predictions of patient drug response. Researchers have developed many ways to build models depending on the type of input data and desired algorithm (reviewed in [7]). We previously established a general approach to impute/predict drug response in patients that was shown to be accurate in retrospective analyses of clinical studies [8]. Our method involves building predictive models between baseline gene expression values in cell lines and their respective drug efficacy metrics (e.g., EC50 or AUC). In our original publication, this modeling approach was shown to be equally good or better at predicting patient response as the gene signatures derived directly from the clinical datasets. This retrospective study and others [9,10] have shown that our methodology is accurate and useful for identifying meaningful relationships between drug response and patient populations.

Most advancement in drug imputation has focused on improving modeling methods so that we can better stratify “responder” from “non-responder.” Recently we have begun to investigate extensions beyond obtaining accurate predictions of patient response. For example, we previously linked patient imputed drug response with genomic features and, in doing so, recapitulated known and discovered new biomarkers of drug response [11]. Here we propose a novel use case for patient drug modeling: drug repositioning and utility extension. We hypothesize that we can flip the traditional paradigm of patient drug response modeling in order to identify drugs targeted towards a particular patient population. That is, instead of stratifying patients into responder/non-responder populations, we could begin with the patient population we would like to respond and test for compounds predicted to target this patient subset. Overall, we contend that drug sensitivity prediction methods can fill in the often-missing pharmacological data from clinical patient datasets, providing a virtual drug screen of patients to hundreds of compounds and allowing for the identification of trends among imputed drug response, clinical features, and patient subtypes.

For this study, we aimed to identify compounds that could lead to the more effective treatment of TNBC. We present our findings in two phases: the discovery phase and the validation phase. Using a candidate drug as an example, we demonstrate the process of identifying a lead candidate drug and perform biomarker discovery for the drug of interest. Then, we validate the method using an independent cell line drug screening dataset and use in vitro and in vivo experiments to explore the utility of the candidate drug with existing standard of care treatment.

## 2. Materials and Methods

### 2.1. Data Acquisition and Code Availability

The Broad Institute’s Cell Therapeutics Response Portal v2 (CTRP) [12] AUC data were obtained from the Cancer Target Discovery and Development Network established by the National Cancer Institute’s Office of Cancer Genomics [13]. The corresponding gene expression values for these cell lines were obtained directly from the Broad Institute’s Cancer Cell Line Encyclopedia (CCLE) data portal [14]. The Cancer Genome Atlas (TCGA) [15] gene expression data was downloaded from firebrowse.org and the clinical data (IHC status, PAM50 subtype, etc.) was downloaded using the TCGAbiolinks R package [16]. The Genomics of Drug Sensitivity in Cancer (GDSC) [17] data was downloaded the GDSC website [18].

R Code to reproduce the entire computational analysis is available from the Github repository at (https://github.com/RFGruener/Gruener-et-al_2021). There, R scripts and additional documentation allows for the download of the CCLE, CTRP, GDSC, and TCGA data, CTRP/CCLE model generation and imputation in TCGA, statistical analyses to identify compounds of interest, and biomarker analysis.

### 2.2. Generating Models for Imputing Drug Response and Statistical Analysis

The methods for imputing drug response in TCGA patients using the CTRP/CCLE cell line data are based on those previously described [8]. To summarize the methodology here, TCGA and CCLE expression data were filtered for common genes between the two dataset and then integrated using ComBat [19]. Feature selection was performed by removing 20% of genes with the lowest variation in gene expression across the samples. After a power transformation of the AUC values, a linear ridge regression model was fit between the CCLE gene expression and corresponding cell line AUC values from CTRP for every drug independently. Once the models were fit, we input the homogenized TCGA patient gene expression data into the models to obtain a drug sensitivity estimate (imputed sensitivity score) for each patient to every drug in CTRP.

### 2.3. Criteria for Lead Compound Identifcation and Statistical Analysis

Patients were grouped into clinical or PAM50 subtypes and the imputed sensitivity scores for each patient were compared using a two-sided Welch Two Sample t-test. For the proof-of-concept comparisons for drugs effective in the hormone receptor-positive (HR+) setting, patients were separated into HR+ and TNBC, and t-tests were performed on the respective imputed sensitivity scores. For comparisons looking for drugs effective in the TNBC setting, patients were stratified into TNBC and non-TNBC groups and t-tests were performed on the respective imputed sensitivity scores. Given the large sample size (*n* > 1000) for these t-tests, the number of significant associations and degree of the significance could be quite high even after multiple-test corrections. This enabled us to be stricter in our criteria for compound-of-interest identification. For the HR+ analysis, only the 10% most significant compounds predicted to be more effective in the HR+ subset were investigated further. For the TNBC analysis, similar criteria were employed, selecting discoveries based on both a top 10% significance and a top 10% effect size thresholds. This second criterium was added because the effect size values were in general skewed towards TNBC for biologically unspecific reasons, as mentioned in the discussion. T-tests, *p*-value adjustments, and Spearman’s correlation tests were performed using the base functions in R. Data was graphed using the package ggplot2 [20]. Mechanism of action, target information, and clinical phase were obtained from a recent review [21]. 

### 2.4. Gene-Set Enrichment Analysis

Gene-set enrichment analysis [22] was performed using the software package GSEA v4.0.2 for Windows downloaded from gsea-msigdb.org. TCGA BRCA RNA-Seq data was used as the expression dataset, MSigDB’s hallmark gene sets [23] were used for the gene sets database, and patient imputed sensitivity scores to AZD-1775 were used as a continuous phenotype label. Default software parameters were used except Pearson correlations were used for ranking genes to reflect the use of a continuous phenotype label. 

### 2.5. Obtaining Biomarker Associations between Imputed Drug Response and Nonsynonymous Somatic Mutations and GDSC ANOVA Biomarker Associations

The associations between imputed drug response in TCGA and somatic mutations were calculated using linear models in R as previously described [11]. Briefly, gene mutation information was obtained from firebrowse.org (2016/01/28 release), which were summarized at a gene level with mutations called if a gene contained a nucleotide change that would affect the protein’s amino acid sequence. We controlled for cancer type when the analysis was applied to all TCGA or PAM50 subtype for the TCGA BRCA cohort when specified in the text by including cancer type/subtype as a covariate (encoded as a factor) in the linear models. 

ANOVA associations between drug response and TP53 for all 185 drugs in the GDSC2 dataset was downloaded directly from the GDSC data portal [24]. The *p*-values were FDR corrected for the 185 associations tested.

### 2.6. In Vitro Cell Line Experiments

BT549, HS578T, and MDA-MB-231 cell lines (ATCC) were maintained in RPMI (ThermoFisher Scientific, Waltham, MA, USA), DMEM, and DMEM (GE Healthcare Life Sciences, Hyclone, Logan, UT, USA) media respectively. All media was supplemented with L-glutamine and 10% FBS (ThermoFisher Scientific, Gibco, Waltham, MA, USA). For viability assay, cells were seeded at 5000 cells/well in 96-well plates. After 24 h, the media was removed and replaced with media containing AZD-1775 at various concentrations between 0 and 3.2 µM, DMSO was used as vehicle and given in control wells. Growth was monitored every 4 h to ensure control wells reached but did not exceed 95% confluence. After approximately 72 h of treatment for each cell line, Cell Titer Glo^®^ (Promega, Madison, WI, USA) viability assay was performed as suggested by manufacturer. Luminescence values were obtained from VICTOR Multilabel plate reader (PerkinElmer, Waltham, MA, USA) and normalized to control well before plotting. Graphing and IC50 determinations were done using Prism 8 software (GraphPad, San Diego, CA, USA).

### 2.7. Xenograft Experiments

All mouse studies were performed under the approved Institutional Animal Care & Use Committee (IACUC) protocol number 72037. C.B17 SCID mice were purchased from Taconic Biosciences. At 8 weeks of age, 3 × 10^6^ MDA-MB-231 cells in PBS and Matrigel^TM^ (Corning, New York, NY, USA) were injected into the mammary fat pads of the mice. When average tumor size reached 150 mm^3^, mice were randomized into 4 treatment groups including vehicle and combination. AZD-1775 was received from AstraZeneca through the NIH’s CTEP program, prepared in 0.5% methylcellulose solution, and delivered via oral gavage at 75 mg/kg on the first three days of the week for 4 consecutive weeks. Doses and schedule of AZD-1775 were suggested by AstraZeneca in order to best mirror use in patients. Paclitaxel from Teva Pharmaceutical (NDC 1703-4768-01) was purchased from the University of Chicago Pharmacy, prepared in PBS, and delivered by IP injection at 12 mg/kg on the first day of the week for 4 consecutive weeks. Tumor volume was monitored twice weekly by caliper and measured using the formula π/6 × L × W^2^. Survival analyses are based on when tumors reached a study endpoint of 2000 mm^3^. Graphing and statistical analyses performed using Prism 8 software (GraphPad, San Diego, CA, USA).

## 3. Results

### 3.1. Discovery Phase: Imputing Patient Response to Medications Enables the Discovery of Candidate Drugs for TNBC

CTRP [12] is the largest publicly available cancer cell line screening dataset with 496 unique compounds screened in 887 cancer cell lines. We used CTRP’s publicly available in vitro drug response data and the corresponding RNA-Seq gene expression data from CCLE [25] for model building as described in the methods. Because each model is generated independently, the accuracy of these models can vary. Therefore, we performed a 20-fold cross-validation, and, of the original 496, only the 427 drug response models that had both a significant and positive Spearman correlation between measured and predicted response were further examined for the rest of this paper (Appendix A). We then applied these drug response models to the breast tumor RNA-Seq data from TCGA [26] to obtain a drug sensitivity estimate for each drug against each patient (Appendix A).

In order to discover drugs that are targeted towards TNBC, patients were stratified based on tumor IHC status for ER, PR and HER2 and patterns of imputed drug sensitivity in each subtype were compared. As a proof-of-concept, we first sought out drugs that were predicted to be targeted towards hormone receptor-positive (HR+, i.e., ER+ and/or PR+) breast cancers. By stratifying patients by their HR-positivity and comparing patient imputed drug sensitivities, we identified 11 compounds predicted to be preferentially effective in HR+ cancers (Appendix A). The two most significant results were a BCL2 inhibitor and tamoxifen, the standard-of-care ER antagonist. BCL-2 is overexpressed in 80% of ER+ cancers and inhibitors have already been investigated for HR+ cancers in clinical trials [27]. These results were encouraging and suggested that this approach could indeed identify relevant compounds-of-interest for a patient population.

We employed the same approach to identify compounds effective for TNBC patients. We dichotomized patients into TNBC and receptor positive (ER+, PR+, or HER2+; abbreviated RPBC) categories and looked for compounds that showed greater predicted efficacy in TNBC compared to RPBCs by t-test (Appendix A). Figure 1A shows a volcano plot of the t-test results for all 427 drugs investigated. Due to the large sample size (*n_total_* = 1095), 251 drugs showed higher sensitivity in TNBC when compared to RPBC at a Bonferroni adjusted *p*-value of less than 0.01 (see discussion). Thus, we chose to enforce a stricter threshold and focused on the top 10% of compounds showing higher predicted efficacy in TNBC based on both effect size and statistical significance, the results of which can be seen in Table 1. Of note, the most frequently represented pathway targeted by these compounds was cell cycle related (12 out of 17 drugs of interests; mechanism of action starred in Table 1). This agrees with previous studies that have identified the cell cycle as a vulnerability in TNBC [28]. Furthermore, all ten pathways identified by this analysis have been implicated as dysregulated in TNBC. Several of the candidate compound nominated by our approach have already been investigated in preclinical or clinical settings. References to the preclinical and clinical investigation of these compounds in TNBC can be found in Appendix A. Overall, our approach appears consonant with and in support of the more traditional approaches that led to the identification of these drugs for TNBC in the literature; thus, substantiating the accuracy of our results and the potential of our computational approach to help speed up the drug selection pipeline.

Of the most significant hits, the top two compounds—AZD7762 and leptomycin B—have been studied in clinical trials in cancer. However, the development of these two compounds was halted due to toxicities. The third most-significant hit, AZD-1775 (aka MK-1775 and Advosertib), was well tolerated in patients in a phase I clinical trial in advanced solid tumors [29]. AZD-1775 also had a high in vitro cross-validation score (r_s_: 0.55, *p*-value 3.3 × 10^−60^) and targets the cell cycle like many of the other most significant results. Finally, AZD-1775 was also consistently one of the most significant results based on both clinical subtypes based on receptor status (Figure 1B) and PAM50 subtype classifications (Figure 1C). For these reasons, we chose to focus on AZD-1775 for subsequent validation and to demonstrate the potential/feasibility of our drug selection/validation pipeline.

### 3.2. Discovery Phase: Identify Biomarkers for AZD-1775

#### 3.2.1. Proof-of-Concept: Tumors Predicted to Be Sensitive to AZD1775 Are Enriched with Cell Cycle Gene Sets

The primary target of AZD-1775 is the Wee1 kinase, which is known to play a critical role in inhibiting the cell cycle at the G2/M checkpoint. We hypothesized that, if our model is picking up on biological meaningful patterns, the RNA expression profiles of patient tumors predicted to be more sensitive to the AZD-1775 should be enriched for cell cycle gene sets. To test this hypothesis, we performed gene-set enrichment analysis (GSEA) on the TCGA breast cancer RNA-seq data using patient imputed response to AZD-1775 as the continuous phenotype label. Using the hallmark gene set, we found that tumors predicted to be more sensitive to AZD-1775 were enriched for the G2/M checkpoint signature (Figure 2A, FDR = 0.04, Appendix A). G2/M is the most highly significantly enriched pathway that associated with AZD-1775 sensitivity, indicating a specific and significant concordance between the imputed results and the biological action of AZD-1775.

#### 3.2.2. Imputation-Based Drug-Wide Association Analysis Reveals Potential Biomarkers for AZD-1775

We previously published a method to form associations between imputed drug responses and genomic features in a manner analogous to genome-wide association studies (GWAS) [11]. We employed this methodology (termed IDWAS for imputed-drug wide association study) to link genomic features with our imputed drug response to search for potential biomarkers of response to AZD-1775. Results from this analysis identified mutation status of 13 genes in the TCGA breast cancer cohort that significantly associated with AZD-1775 sensitivity at an FDR < 0.05 (Appendix A). In particular, AZD-1775 response formed a highly significant (FDR = 5.6 × 10^−46^) association with *TP53* mutational status in breast cancer patients (Figure 2B). This p53-association holds when IDWAS was performed on the pan-cancer TCGA with tissue type correction (*n* = 9968, FDR = 2.1 × 10^−8^) as well as in TCGA breast cancer with correction for PAM50 subtype (*n* = 1098, FDR = 1.6 × 10^−9^). The effect size of this association indicates that AZD-1775 would be more effective in the p53 mutated/null setting, which is consistent with other literature that shows Wee1 inhibition in p53 mutated cell lines increases genomic instability and in turn forces apoptosis or mitotic catastrophe [30]. We also investigated the genomic association data in the independent GDSC cell line screening dataset. Of the 185 drugs tested in this data, AZD-1775 was the only drug that significantly associated with increased efficacy in the TP53-mutated setting after multiple test correction (FDR = 0.035, Appendix A), giving further confidence in the association between TP53 mutation and AZD-1775 activity.

To evaluate whether the biomarkers were unique to AZD-1775, we performed the same IDWAS analysis with all 17 drugs from Table 1. We identified 38 genes that associated with these 17 compounds (Appendix A). Six gene mutations (e.g., *TP53*, *PI3KCA*, *MAP3K1*, and *NOTCH3*) associated with over half of all the drugs. This overlap is expected, as many of these drugs have similar mechanisms of action and are imputed to generally work better in the TNBC population. For example, the 8 genes that significantly associated with the PLK inhibitor GSK461364 also significantly associated with the other two PLK inhibitors.

Of the 13 significant gene-drug associations with AZD-1775, six genes were found to associate with four or fewer other drugs and one gene associated specifically with AZD-1775 (Appendix A). Of interest were the associations with *ANKRD30A*, *SPTBN1*, and *GALNTL5*, as these genes have known and well-defined cancer associations. *ANKRD30A* encodes for a breast differentiation antigen, NY-BR-1, which is often dysregulated in breast cancer and has been investigated as a general breast cancer biomarker previously [31]. *SPTBN1* and *GALNTL5* have also been implicated in breast cancer. Mutations in these genes have been associated with increase in metastatic phenotypes, which is of interest because both IDWAS associations indicate AZD-1775 would be more effective in the mutated (i.e., more metastatic) setting [32,33].

Overall, we identified a number of candidate drugs and potential corresponding biomarkers in the discovery phase. Next, we shifted to validation of both our ability to generate accurate predictions with AZD-1775 and AZD-1775′s efficacy in TNBC preclinical models.

### 3.3. Validation Phase: Measured Cell Line Response to AZD-1775 in an Independent In Vitro Dataset Validate Our Predictions

Even though the cross-validation within the CTRP dataset showed a good reproducibility between imputed and actual measured AZD-1775 sensitivity, we were interested in further exploring prediction accuracy with another independent dataset. Therefore, we imputed drug response scores in GDSC and compared GDSC measured AZD-1775 IC50 to our imputed results. We find a significant positive Spearman-rank correlation between our imputed sensitivity scores and GDSC’s available IC50 data (Appendix A, *n* = 749, r_s_ = 0.63, *p* < 2.2 × 10^−16^). This correlation between imputed and measured sensitivity remains significant when we only analyze cell lines unique to GDSC (Figure 3A, *n* = 297, r_s_ = 0.62, *p* < 2.2 × 10^−16^). For comparison, the Spearman-rank correlation between AZD-1775 sensitivity in the cell lines that were measured in both datasets was 0.50 (overlapping cell lines: *n* = 452, r_s_ = 0.50, *p* < 2.2 × 10^−16^, Figure 3B). That the correlation of the imputed and measured results in GDSC was comparable to (if not greater than) the correlation between measured GDSC and CTRP results indicates that our imputed values capture the majority of AZD-1775′s in vitro sensitivity variation. A breast cancer cell line only analysis reveals a similar correlation between imputed AZD-1775 response and GDSC measured sensitivity (Appendix A); as did an analysis using only breast cancer cell lines unique to GDSC (Appendix A). The correlation between the measured response in the 33 breast cancer cell lines present in both CTRP and GDSC was 0.64 (Appendix A).

### 3.4. Validation Phase: In Vitro and In Vivo Assessment of Cellular Sensitivity to AZD-1775 in Combination with Standard-of-Care Paclitaxel

Our overall goal is to identify compounds that will be effective in the treatment of breast cancer patients. While the differences in imputed response indicate that a druggable vulnerability exists between the two populations, this data does not prove that the drug will be potent/effective at the appropriate dose levels. Thus, it is of paramount importance to follow-up the lead-compound identification with tests to verify a drug’s potency in the population of interest. This section addresses this by looking at AZD-1775′s inhibitory ability in in vitro and in vivo models of TNBC.

#### 3.4.1. Single Agent use of AZD-1775 Is Able to Inhibit Growth of TNBC Cell Lines

Thus far, we have focused mainly on CTRP data, which, as a high-throughput screen, is not optimized for particular compounds or cell lines. For example, AZD-1775 was screened at a concentration range between 0–33 µM while the reported clinical *C_max_* is only 1.4 to 2.6 µM depending on the dosing schedule [34]. We therefore performed cell viability assays (Cell Titer Glo^®^, Promega, Madison, WI, USA) tailored to three TNBC cell lines at a more pharmacologically meaningful dose range of 0 to 3.2 µM. We found EC50 to be 392.8 nM, 531.2 nM and 602.5 nM in the BT549, HS578T and MDA-MB-231 cell lines respectively after 72 h of AZD-1775 treatment (Appendix A). These results are in line with CTRP’s reported EC50 values, where 18/19 TNBC cell lines were found sensitive to AZD-1775 at an EC50 1.4 µM or less (Appendix A). These findings support that AZD-1775 is efficacious in inhibiting the proliferation of TNBC cell lines at pharmacologically achievable concentrations.

#### 3.4.2. AZD-1775 Alone and in Combination with Paclitaxel Inhibits MDA-MB-231 Xenograft Growth

To further validate AZD-1775 for use in TNBC, we performed tumor growth experiments in a mouse xenograft model. Using orthotopic MDA-MB-231 xenografts, we tested single agent AZD-1775 compared to vehicle control and/or paclitaxel, a standard of care agent in treating TNBC, as a positive control. Regarding tumor outgrowth, single-agent paclitaxel and AZD-1775 had similar efficacy at most time points, but neither showed consistent statistically different results when compared to control after adjusting for multiple comparisons (Figure 4A, AZD-1775 alone only reached significance at day 10, adj. *p* = 0.0491). However, time-to-endpoint analysis showed that AZD-1775 significantly improved time to endpoint while paclitaxel did not (Figure 4B, AZD-1775: *p* = 0.021; paclitaxel: *p* = 0.18 by Log-rank test). Additionally, we tested AZD-1775 in combination with paclitaxel; the rationale being that novel drug candidates are often first evaluated in clinical trials in combination with a standard-of-care agent. Combination treatment with AZD-1775 and paclitaxel showed statistically significant impairment of tumor outgrowth by day 10, which remained significant over the treatment period (Figure 4A, multiplicity adjusted *p*-value <0.05). Mice receiving a combination of AZD-1775 and paclitaxel did significantly better in time-to-endpoint than both the control (*p* = 0.0003) and paclitaxel alone (*p* = 0.017). These results support the use of AZD-1775 as a monotherapy, but they more strongly support the future testing of the combination of paclitaxel and AZD-1775 in treating TNBC.

## 4. Discussion

TNBC is a very aggressive disease with a high recurrence rate and limited therapeutic options [35]. Our main goal was to take advantage of existing large high-throughput in vitro drug screening datasets and apply novel computational methods to systematically identify effective drugs for the treatment of TNBC. To this end, we imputed drug responses by making 427 individual drug-transcriptome models and identified candidate drugs that are likely to be more effective in TNBC tumors. Among them, we focused on AZD-1775, a Wee1 kinase inhibitor, testing the model’s accuracy in an independent screening dataset (GDSC) and the compound’s efficacy in inhibiting tumor growth using an in vivo xenograft model. We also demonstrated the pipeline of employing computational methods to identify biomarkers for compounds of interest that could help further tailor a targeted therapy to the appropriate patient population.

Overall, our method has several unique advantages in comparison with other approaches for drug discovery. First, we are able to investigate a large number of compounds in a relatively short amount of time by taking advantage of existing and publicly available datasets (available databases reviewed in [21]). This analysis was able to quickly query 427 compounds targeting over 400 known cancer targets in a clinical population of interest. Additionally, utilizing machine learning models trained on large panels of in vitro cell line data allows us to make unbiased models with no assumptions about the compounds’ modes of actions, which can often be partially or completely unknown. For cancer indications with limited number of cell lines or useful preclinical models, this method could be particularly useful as the patterns of drug response are identified using large panels of pan-cancer cell lines and then applied to the disease type of interest. Furthermore, the choice of patient population is flexible and controlled by the investigator such that this approach could be used to identify compounds targeted towards patient groups defined clinically or, for example, patients defined by a genetic alteration of interest to the investigator. Finally, imputing in patient datasets directly allows us to translate the in vitro patterns of drug response into the most clinically relevant population and leverage other types of information contained in these datasets. This is something that up until this point has not been done for drug discovery, and, as shown in this work, is quite capable of identifying relevant and useful compounds.

This approach has potential limitations and considerations. This and other drug modeling methods are unable to make absolute potency or efficacy predictions. As such, we can only compare relative imputed sensitivity metrics within a drug and not the metrics between drugs. Issues related to differences in relevant dosing ranges, toxicities, and other pharmacokinetic parameters make comparing drug sensitivity metrics among compounds impractical. In addition, the transformation of sensitivity metrics for modeling and independence of the models add additional barriers that prevent the direct comparison of model outputs between drugs. However, we would argue that even if predicting absolute potency of a compound were possible such that these values were comparable across drugs, it would necessarily enrich for toxic chemotherapies since these are the most potent compounds in general.

Our approach of comparing patient subtypes is therefore more useful for identifying agents that target particular genomic vulnerabilities in the patient population of interest. This does preclude finding compounds that may target vulnerabilities present in both populations at similar frequencies. Under the assumption that a targeted therapy will work for a certain subtype and not others, this limitation is not likely to hinder identifying the most interesting compounds. Indeed, finding targeted therapies and therapies that work specifically for a particular subtype is of critical interest because there is no panacea or magic bullet in cancer. TNBC is a good example use-case then, as TNBC is generally quite distinct from other breast cancer subtypes and indeed in pan-cancer analyses often cluster away from other breast cancers [36,37]. The choice of patient populations to compare should be carefully considered when using this approach.

Of course, both because the method is computationally driven and because predictions cannot check drug potency or toxicities, validation is necessary to check the effectiveness of the compound. As such this method is more complimentary to the traditional drug development pipeline than a substitute for it, allowing for the speedy identification of lead compounds for further study and validation.

The results of the differential imputed drug sensitivity between TCGA’s TNBC and RPBC cohorts indicate that there is a skew towards identifying more compounds predicted to be effective in TNBC rather than RPBC. There are likely several contributing factors to explain this skew. First, regarding the translational models, it has been shown that well performing computational models are not necessarily specific to a drug of interest [38]. That is, general mechanisms of drug response, such as factors associated with multi-drug resistance, are likely encoded in many of the independent models and as such might skew the imputed distribution of all the drugs in a similar direction. Additionally, many of the agents tested in CTRP target DNA-damage or cell cycle, which is a known vulnerability of TNBC, whereas very few compounds target hormone pathways present in RPBC. Finally, RPBC captures a mixture of hormone dependent and growth factor (HER2) driven diseases, thus it is not surprising that not many drugs are specifically enriched in this mixed population. However, for our list of candidate compounds, we only focused on the top 10% most significant hits to account for this skew towards TNBC.

Of the 17 compounds we focused on, we observed multiple cell cycle inhibitors and DNA-damaging agents. Of note were the CHK, PLK, and CDK inhibitors that have been explored in depth in the TNBC literature. CHK inhibitors have already shown promise in TNBC preclinical models to induce cell death by promoting DNA damage and subsequent apoptosis [39,40], with particular efficacy in both Rb [41] and p53 deficient [42] settings. While one of our top results here, the use of CHK inhibitors have been limited due to poor pharmacokinetic properties, with several earlier formulations being discontinued due to toxicities [43]. PLK inhibitors have similarly been explored as an overexpressed factor in TNBC whose inhibition can induce G2/M arrest in TNBC in vitro models [44]. Recapitulating these targets in TNBC supports the ability of our methodology to systematically identify effective agents for a cancer population of interest.

There has been increasing interest in the use of AZD-1775 in various cancer settings, with several early phase clinical trials showing a favorable toxicity profile in a number of different cancers [29,34,45,46,47,48]. Wee1 is a protein kinase that acts as an inhibitor of the cell cycle through phosphorylation of the CDK1/cyclin B complex resulting in G2 cell cycle arrest [49]. Thus, inhibition of Wee1 by AZD-1775 blocks DNA damage repair through loss of the cell cycle checkpoint and promotes early entry into mitosis. This is often fatal for cells, especially in p53 mutated cells that mainly rely on the G2/M checkpoint for DNA repair [50], which is often the case in TNBC [28]. Wee1 inhibition has been shown to sensitize p53-deficient tumor cells to DNA-damaging agents in various cancer contexts [51,52,53,54,55,56,57,58] and to prolong mitosis in breast cancer cells [59]. Since we began investigating AZD-1775 for TNBC, several studies have also come out in TNBC: one showing that cyclin E overexpression sensitizes TNBC to AZD-1775 [60] and one reporting synergy between AZD-1775 and ATR inhibition in TNBC cells [61]. Our work here serves to complement these studies as the first computational approach to use existing pharmacologic data to identify AZD-1775 as a compound of interest for TNBCs as well as the first to explore the combination of AZD-1775 with standard-of-care paclitaxel in TNBC treatment. Of the over 50 clinical trials listed on clinicaltrials.gov with AZD-1775, only one is in (metastatic) TNBC (NCT03012477) and four focus on tumors harboring p53 mutations (NCT02272790, NCT01357161, NCT01164995, NCT02448329). These clinical trials thus far have not provided genomic mutational information to assess the potential of the associated biomarkers we identified. We believe our findings, along with others mentioned here, support further investigation of AZD-1775 in TNBC as well as the investigation of the potential biomarkers identified to tailor therapy to the appropriate patients.

## 5. Conclusions

In summary, we provide a novel approach for the identification and prioritization of candidate compounds for a particular patient group of interest and showcase the use of this method in TNBC. In doing so, we identified and validated AZD-1775 for use in TNBC both in vitro and in vivo. The employment of these computational tools is quick, efficient, and allows for contextualizing cell line response in any patient population. We hope that others in the field are able to make use of this framework for generating drug discovery hypotheses for the patient populations they study and that this ultimately enables the creation of new and effective treatment options for patients.

## Figures and Tables

**Figure 1 cancers-13-00885-f001:**
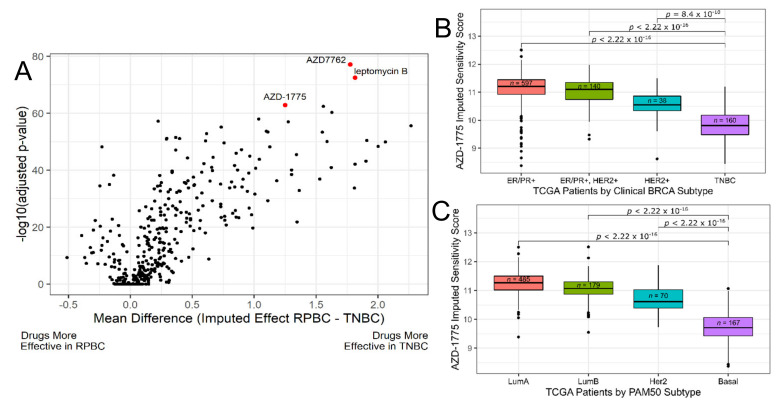
AZD-1775 is predicted to be most effective in TCGA triple-negative breast cancer (TNBC) patients. (**A**) Volcano plot of TNBC vs. receptor positive breast cancer (RPBC, i.e., non-TNBC) imputed sensitivity t-test results for all drugs in CTRP. 427 drug response models were applied to the TCGA breast cancer cohort resulting in an imputed sensitivity score for each patient. A t-test was then performed for every compound between the compound’s imputed response in TNBC and non-TNBC (RPBC) patients. The *p*-values were Bonferroni-adjusted to correct for multiple testing. Highlighted in red are the top 3 most significant results (AZD7762, leptomycin B, and AZD-1775). (**B**,**C**) AZD-1775 imputed sensitivity in TCGA breast cancer tumors by receptor status (**B**) and PAM50 subtyping (**C**). Boxplots summarize results of each tumor sample’s imputed sensitivity score to AZD-1775 in the TCGA breast cancer cohort by subtype. The *n* values indicate the number of patients in each group and *p*-values shown are adjusted for multiple testing. Lower values on the y-axis indicate increased predicted sensitivity. Dataset Abbr: TCGA, The Cancer Genome Atlas; CTRP, Cell Therapeutics Response Portal.

**Figure 2 cancers-13-00885-f002:**
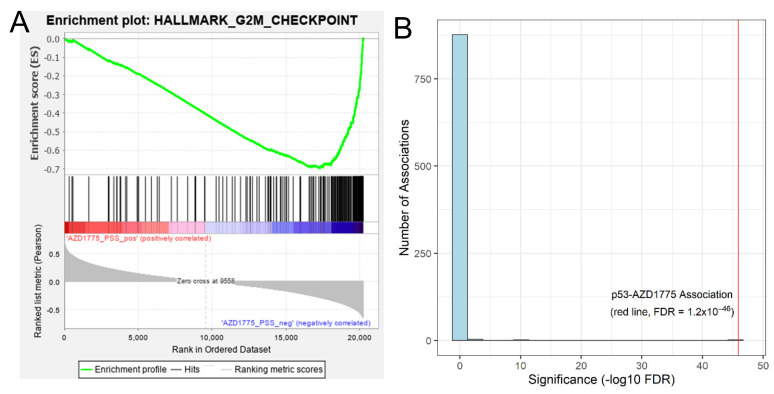
Biologically meaningful associations with AZD-1775 imputed sensitivity in TCGA breast cancers. (**A**) Gene set enrichment analysis was performed in MSigDB’s “Hallmark” gene sets using TCGA’s breast cancer expression data with AZD-1775 imputed sensitivity score as the continuous phenotype variable for enrichment. The G2M gene set was significantly up-regulated in breast cancer patients predicted to be sensitive to AZD-1775 with an enrichment score of −0.695, normalized enrichment score of −1.86 and an FDR q-value of 0.04 (associated data in Appendix A). A negative enrichment score associates with sensitivity to the drug as smaller imputed sensitivity values indicate more sensitive (**B**) A histogram of *p*-values achieved for all the associations between AZD-1775 imputed response and any gene with a somatic protein-coding change in at least 20 samples (*n* = 882 genes) in TCGA breast cancer cohort. *TP53* mutation and AZD-1775 achieves the strongest association at an FDR = 1.2 × 10^−46^, with the next most significant association at an FDR of 1.2 × 10^−13^.

**Figure 3 cancers-13-00885-f003:**
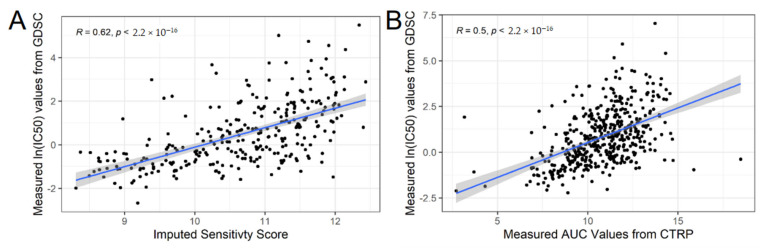
Correlation of Wee1 inhibitor predicted and actual cell line response to AZD-1775 in an independent in vitro dataset. (**A**) For each cell line in GDSC not present in CTRP (i.e., cell lines unique to GDSC), imputed drug response to AZD-1775 using the CTRP imputation model (x-axis) is graphed against the reported GDSC IC50 sensitivities (y-axis). Spearman-rank correlation (*R*) and *p*-value as reported. (**B**) For each cell line in common between GDSC and CTRP, the measured IC_50_ values from GDSC (y-axis) is graphed against to the measured AUC values from CTRP (AUC values are used to build the imputation model). Spearman-rank correlation and *p*-value as indicated. (**B**) serves as a point of comparison for how consistent two completely independent measured datasets are compared to the consistency of our imputed values built in one dataset to the measured values of the other (**A**). Abbr: GDSC, Genomics of Drug Sensitivity in Cancer; IC50, Half maximal inhibitory concentration; AUC, Area Under the dose response Curve.

**Figure 4 cancers-13-00885-f004:**
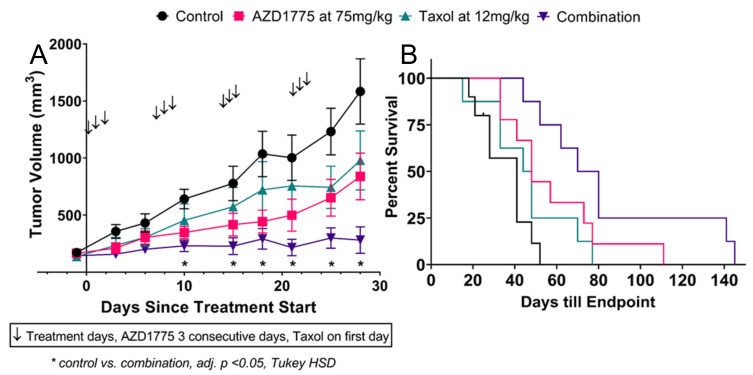
AZD1775 and paclitaxel combination inhibits TNBC tumor growth in vivo. (**A**) Tumor growth assay of MDA-MB-231 xenografts treated with AZD1775 and paclitaxel. Mice were orthotopically injected with MDA-MB-231 cells and allowed to grow to an average tumor size of 150 mm^3^ before being randomized into 4 treatment groups: vehicle control, AZD-1775 given PO at 75 mg/kg on consecutive days 1–3 of each week for 4 weeks, Paclitaxel bolus given IP every week on day 1 of each week for 4 weeks, and the combination of the paclitaxel and AZD-1775 treatments. Tumor size was measured by caliper every 3–4 days after treatment start and tumor volumes and standard errors are plotted by treatment group over the first four treatment weeks. * indicate significant values between combination and control by Tukey’s honestly significant difference (HSD) for each timepoint. The only other value that was significantly different during this time period by HSD was the AZD-1775 compared to control treatment at day 10 (not indicated). (**B**) Days till endpoint analysis of MDA-MB-231 xenografts. For the same mouse experiment shown in (**A**), days until tumor growth endpoint (2000 mm^3^) were recorded and plotted as a Kaplan–Meier survival curve. Log-rank (Mantel–Cox) test performed for all 4 curves reached a significance of *p* = 0.0014. Individual analysis was performed between each curve, the log-rank test was only significant (*p* < 0.05) when comparing Control vs. combination, *p* = 0.0003, Taxol vs combination, *p* = 0.0086, and Control vs. AZD1775, *p* = 0.0207.

**Table 1 cancers-13-00885-t001:** Drug pathways predicted to be effective for TNBC based on the differential imputed response analysis. Table 1 contains the mechanisms of action (MOA) of the compounds that were in the top 10% most effective for TNBC based on both effect size and *p*-value from the imputed sensitivity analysis in the TCGA breast cancers. The rows are in ordered from most to least significant based on t-test *p*-value of the first drug listed in the drug column. A count column for the total number of drugs with the same MOA are also included. Asterisk (*) in Mechanism of Action column indicate drugs that target cell cycle/DNA repair pathways.

Mechanism of Action	# of Drugs in Top 10%	Total # of Drugs in Database	Drug(s) in Top 10%
CHK inhibitor *	1	1	AZD7762
exportin antagonist	1	1	leptomycin B
WEE1 kinase inhibitor *	1	1	AZD-1775
CDK inhibitor *	5	6	dinaciclib, alvocidib, SNS-032, PHA-793887, BRD-K30748066
translation (eIF4F complex) inhibitor	2	2	CR-1-31B, SR-II-138A
PLK inhibitor *	3	4	GSK461364, BI-2536, rigosertib
proteasome inhibitor	1	2	MLN2238
tubulin polymerization inhibitor *	1	4	docetaxel
phosphodiesterase inhibitor	1	2	ML030
kinesin-like spindle protein inhibitor *	1	1	SB-743921

## Data Availability

All data from CTRP, CCLE, GDSC, and TCGA are publicly available as listed in the methods. All code needed to reproduce the results from this paper can be found at the Github Repository: https://github.com/RFGruener/Gruener-et-al_2021.

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
