# Peer review of "Facilitating Drug Discovery in Breast Cancer by Virtually Screening Patients Using In Vitro Drug Response Modeling"

_cancers, 2021, doi:10.3390/cancers13040885_

Round 1

Reviewer 1 Report

This paper used an established machine learning approach to build models of drug response in breast cancer patients. The authors claimed that they had developed a model to identify compounds which effectively treat triple-negative breast cancer. One major question of this research is that the authors are actually identifying compounds showed differential activity between TNBC patients and other breast cancer patients or TNBC specific compounds. However, specificity does not guarantee effectiveness. They could easily miss compounds that can effectively treat TNBC patients and other breast cancer patients. The compounds they identified may be high specificity to TNBC patients but with low effectiveness. Thus the significant of this model is relatively low. Besides that, the model was not clearly presented and key data, sensitivity score, was missing.

  1. The fundamental question is whether the model can predict drug effectiveness or in other words can we compare between different compounds. If compound A get a sensitivity score of 10 and compound B get a sensitivity score of 15, can we say compound A is more effective than compound B? If yes, the author should just focus on the TNBC patients and look at the compounds that have the lowest sensitive score instead of comparing between patients. If not, the significant of the model is relative low and please explain why.
  2. Only showing the sensitivity score of Tamoxifen is not strong enough to support you model. The best way to verify your model is to see if you can successfully identify Tamoxifen as a specific treatment for ER+ breast cancer patient. The author can compare the predicted efficacy between ER+ patients and ER- patients or luminal patients and other subtypes use the same criterial( questioned on 3) and see if tamoxifen is among the most effective compounds.
  3. The author state Table 1 contains Table10 most effective for TNBC based on both effect size and p-value. First, please correct the typo. Second, the screening criterial is not clearly stated. Judging from the SupTable1, it seems the author only ranked the compounds with effective size >1 by P-value. Please explain why p-value was used to rank the compounds instead of effective size. Third, the triazolothiadiazine in Table1 is not included in SupTable1. Is that an alternative name? The author state that it is a ceramidase inhibitor. However, the only ceramidase inhibitor in supTable1 is NVP-231. Fourth, why ML030 is not in Table 1? ML030 has a better P-value than SB-743921, BI-2536, SR-II-138A and rigosertib.
  4. Only triple negative breast cancer cell lines are used in validation. Only 1 cell line is used in validation. Since the author is trying to show specificity, other subtypes of breast cancer cell lines should be used in validation.
  5. As the author state that P53 could be a selective marker, cell lines with different p53 statues should be validated at least in vitro.
  6. Sensitivity score of each compound in each patients should be supplemented as this is the direct and most important data from your model.

Reviewer 2 Report

In this manuscript Gruener et al.,  investigated the use of drug prediction methods for the generation of novel drug discovery hypotheses for new therapies using an established machine learning methods to build models of drug response based on invitro data performed to secure predicted sensitivity scores in over 1000 breast cancer patients.  Triple-negative breast cancer is considered as a very aggressive cancer which grows rapidly with fewer therapeutic options. Author used following methodology to investigate this study: Data Acquisition, Generating Models for  Drug Response and Statistical Analysis, Obtaining Associations Between Imputed Drug Response and Nonsynonymous Somatic Mutations, Gene-Set Enrichment Analysis, In vitro  Experiments using cell lines such as BT549, HS578T, and MDA-MB-231, Xenograft Experiments. Based on the  computational analysis author focused on a wee1 kinase inhibitor AZD-1775,  testing the model’s accuracy in an independent screening dataset  and the compound’s efficacy in inhibiting tumor growth using an xenograft model. Author identified and validated AZD-1775 for use in TNBC both in vitro and in vivo. The  computational and experimental tool in the drug development pipeline could be useful for  identifying other cancer subtypes for generating new compounds of interest for novel drug discovery.

My comments are very positive. The manuscript is well written and the overall data are clearly presented and computational analysis have been carefully executed and shows all the required controls and results contain important novel information.

Author Response

Thank you for the consideration you gave our manuscript. From the summary you provided, it is clear we were able to communicate our work effectively and we appreciate the vote of confidence you gave our work.

Once again, thank you for your time and attention.

All my best,

Dr. R. Stephanie Huang

Round 2

Reviewer 1 Report

the author has addressed most of the concerns and the quality of the manuscript has been improved drastically.